# Zika Virus Immunoglobulin G Seroprevalence among Young Adults Living with HIV or without HIV in Thailand from 1997 to 2017

**DOI:** 10.3390/v14020368

**Published:** 2022-02-10

**Authors:** Sirinath Choyrum, Nantawan Wangsaeng, Anouar Nechba, Nicolas Salvadori, Rumpaiphorn Saisom, Jullapong Achalapong, Chaiwat Putiyanun, Prapan Sabsanong, Suraphan Sangsawang, Orada Patamasingh Na Ayudhaya, Gonzague Jourdain, Nicole Ngo-Giang-Huong, Woottichai Khamduang

**Affiliations:** 1Department of Medical Technology, Faculty of Associated Medical Sciences, Chiang Mai University, Chiang Mai 50200, Thailand; sirinath.pch@gmail.com (S.C.); nicolas.salvadori@phpt.org (N.S.); gonzague.jourdain@phpt.org (G.J.); Nicole.Ngo-Giang-Huong@phpt.org (N.N.-G.-H.); 2Associated Medical Sciences (AMS)-CMU IRD Research Collaboration, Chiang Mai 50200, Thailand; Nantawan.Wangsaeng@phpt.org (N.W.); anouar.nechba@phpt.org (A.N.); rumpaiphorn.saisom@gmail.com (R.S.); 3Maladies Infectieuses et Vecteurs: Écologie, Génétique, Évolution et Contrôle (MIVEGEC), Agropolis University Montpellier, Centre National de la Recherche Scientifique (CNRS), Institut de Recherche Pour le Développement (IRD), 34394 Montpellier, France; 4Chiangrai Prachanukroh Hospital, Chiang Rai 57000, Thailand; jullapong.achalapong@phpt.org; 5Chiang Kham Hospital, Phayao 56110, Thailand; putiyanun@hotmail.com; 6Samutsakhon Hospital, Samutsakhon 74000, Thailand; prapandr@hotmail.com; 7Health Promotion Center Region 1, Chiang Mai 50100, Thailand; suraphan3107@gmail.com; 8Nopparat Rajathanee Hospital, Bangkok 10230, Thailand; oradaaom@hotmail.com

**Keywords:** Zika virus, epidemiology, IgG, seroprevalence, Thailand, HIV, pregnant women

## Abstract

Zika virus (ZIKV) epidemiological data in Thailand are limited. We assessed ZIKV IgG seroprevalence among young adults during 1997–2017 and determined factors associated with ZIKV IgG seropositivity. This retrospective laboratory study included randomly selected subjects aged 18–25 years participating in large clinical studies conducted in Thailand during 1997–2017. Stored plasma samples were analyzed for ZIKV IgG using an ELISA test (Anti-Zika Virus IgG, EUROIMMUN, Lübeck, Germany). Sociodemographic, clinical and laboratory data were used in univariable and multivariable analyses to identify factors associated with ZIKV IgG positivity. Of the 1648 subjects included, 1259 were pregnant women, 844 were living with HIV and 111 were living with HBV. ZIKV IgG seroprevalence was similar among the HIV-infected and -uninfected pregnant women (22.8% vs. 25.8%, *p*-value = 0.335) and was overall stable among the pregnant women, with a 25.2% prevalence. Factors independently associated with ZIKV IgG positivity included an age of 23–25 years as compared to 18–20 years, an HIV RNA load below 3.88 log_10_ copies/mL and birth in regions outside northern Thailand. Our study shows that a large proportion of the population in Thailand probably remains susceptible to ZIKV infection, which could be the ground for future outbreaks. Continued surveillance of ZIKV spread in Thailand is needed to inform public health policies.

## 1. Introduction

Zika virus (ZIKV) is an enveloped, positive single-strand RNA virus belonging to the *Flaviviridae* family and *Flavivirus* genus. The RNA genome is composed of about 10,800 nucleotides encoding three structural proteins, the capsid, precursor membrane and envelope protein, and seven nonstructural (NS) proteins i.e., NS1, NS2A, NS2B, NS3, NS4A, NS4B and NS5 [1]. ZIKV is mainly transmitted to humans through infected *Aedes* mosquito bites. Since its discovery in 1947 in Uganda, ZIKV infection was not considered as a public health concern until the outbreaks in the Pacific region between 2007 and 2013 [2,3,4]. Indeed, an acute infection is usually asymptomatic or exhibits mild and self-limiting symptoms. When present (less than 20%), symptoms include a non-specific febrile syndrome with a maculopapular rash, arthralgia, or conjunctivitis [3,5].

In 2007, the first ZIKV outbreak outside Africa and Asia occurred in Yap Island (Federated States of Micronesia) [3,5]. It was followed in 2013 with an outbreak in French Polynesia, which was responsible for severe neurological complications in adults and malformation in neonates [2,6,7]. The virus subsequently spread to South and Central Americas in 2015 [8], especially in Brazil, where ZIKV infection was associated with neurological complications, including microcephaly in newborns or Guillain–Barré syndrome (GBS) in adults [2,9,10]. As a result of this extensive spread of ZIKV and its associated neurological complications, the World Health Organization designated ZIKV a “Public Health Emergency of International Concern” in February 2016 [11].

Initial serologic tests performed on stored samples suggest that ZIKV has circulated in Thailand since 1954 [12]. In 2013, the Ministry of Public Health of Thailand (Thai-MOPH) rapidly implemented in the local healthcare centers a system to report ZIKV infections following the report of a symptomatic ZIKV infection in a traveler upon returning to Canada after visiting Thailand in May 2013 [13] and the suspicion of a ZIKV outbreak in several areas. Shortly after, the Thai-MOPH conducted ZIKV investigations throughout the country. A retrospective analysis of neutralization antibody in stored plasma samples collected in 2012 from two patients with exanthematous fever identified that they had been infected with ZIKV [14]. However, no outbreaks and no severe complications have ever been reported.

We present herein the seroprevalence of immunoglobulin G (IgG) against ZIKV among young adults in Thailand over several time periods between 1997 and 2017 and factors associated with positive ZIKV IgG.

## 2. Materials and Methods

### 2.1. Study Population

This is a retrospective laboratory study of ZIKV IgG among subjects enrolled between 1997 and 2017 in large clinical studies conducted in Thailand on the prevention of perinatal transmission of HIV [15,16,17,18,19] or hepatitis B virus (HBV) [20] or in an HIV testing research program [21] (ClinicalTrials.gov Identifier: NCT00386230, NCT00398684, NCT00142337, NCT00409591, NCT01511237, NCT01745822, NCT02752152, respectively). Since cumulative exposure to mosquitoes increases over an individual’s lifetime, the risk of being ZIKV-IgG-positive may increase with age. For this reason, only subjects aged 18–25 years were included in this study. For this study, we used socio-demographic and clinical data, laboratory results, and stored blood samples that were collected during the course of those studies.

Of the 8347 subjects enrolled between December 1997 and December 2017 across the seven studies and with a stored sample, 3675 met the age range criterion. Only 97 women enrolled in the perinatal HIV prevention studies conducted during the period 2004–2007. These 97 pregnant women were not included in this Zika study since their number was too low to allow for an appropriate random age-based selection. We, thus, considered five time periods, based on the years in which those studies were conducted: 1997–2000 (742 subjects), 2001–2003 (833 subjects), 2008–2011 (158 subjects), 2012–2014 (238 subjects) and 2015–2017 (1704 subjects). We used a proportionate sampling approach to obtain the target number of subjects for each period, i.e., 400, 250, 150, 150, and 400, respectively (Figure 1). To homogenize the study population, the subjects at each time period were separated into four sub-groups according to the subjects’ age quartiles. We then applied an age-matched draw procedure to select the subjects from each time period.

### 2.2. Laboratory Testing

Stored plasma samples collected before any antiretroviral treatment were tested for ZIKV IgG using an indirect ELISA test (Anti-Zika Virus IgG ELISA, EUROIMMUN, Lübeck, Germany; Product number: EI 2668-9601 G; 78.9% sensitivity and 99.8% specificity [22]) according to the manufacturer’s instructions. Each test run was validated with the kit positive and negative controls as internal controls. A test was considered ZIKV-IgG positive if the signal per cut-off ratio was >1.1 and ZIKV-IgG negative if the ratio was ≤1.1.

### 2.3. Statistical Considerations

The characteristics of the subjects are described using counts and percentages for categorical data and medians with interquartile ranges (IQR) for continuous data. The characteristics included age at enrollment, region of birth, occupation, education level, blood chemistry and hematology tests, HIV, hepatitis B and C virus and syphilis infection statuses and HIV-1 RNA load. The percentage of women with ZIKV IgG antibodies, along with the corresponding Clopper–Pearson 95% confidence interval (CI), are provided for each group. ZIKV IgG seroprevalence during the 1997–2000 period was compared between the HIV-infected and HIV-uninfected pregnant women. ZIKV IgG seroprevalence was analyzed at each of the five time periods and compared to the ZIKV IgG seroprevalence in 1997–2000 using a *chi*-square test.

Logistic regression models were used to identify whether time periods and other factors were associated with ZIKV IgG positivity. Continuous variables were transformed into categorical variables using common cut-off or median values. All factors with a *p*-value < 0.2 in the univariate analysis were considered for inclusion in the multivariate analysis, and the backward elimination procedure was applied to select only independent factors associated with ZIKV IgG positivity. All data analyses were performed using Stata™ version 14.1 software (Statacorp, College Station, TX, USA). Differences were considered statistically significant if the *p*-value was ≤0.05.

## 3. Results

### 3.1. Study Population Characteristics

Of the 1750 randomly selected subjects, 1648 had a sample available (386 in the period 1997–2000, 248 in the period 2001–2003, 102 in period the 2008–2011, 113 in period the 2012–2014, 399 in the period 2015–2017 and 400 in the HIV-uninfected pregnant women) (Figure 1). The median age was 22 years (IQR: 20–23 years). Of the 1648 subjects with samples available, 1464 (88.8%) were females, of whom 1295 were pregnant, with a median gestational age of 25 weeks at time of blood draw. Almost half of subjects were born in the northern region of Thailand (Table 1), and 956 (58.4%) completed secondary or higher education.

Eight hundred and forty-four subjects (51.2%) were positive for HIV antibodies. The median HIV-1 RNA load was 3.88 log_10_ copies/mL (IQR: 3.21–4.46). The median CD4 T-cell count was 410 cells/mm^3^ (IQR: 280–550), and 13.5% (110 of 814) had a CD4 T-cell count below 200 cells/mm^3^. The hepatitis B surface antigen was positive in 111 of 1245 subjects (8.9%) and the HCV antibody was positive in 22 of 1246 (1.8%). Other socio-demographic data, laboratory test results and substance use information are described in Table 1 and Appendix A.

### 3.2. ZIKV IgG Seroprevalence in HIV-Infected versus HIV-Uninfected Pregnant Women during 1997–2000

During 1997–2000, 88 of 386 (22.8%, 95%CI: 18.7–27.3) HIV-infected pregnant women tested positive for ZIKV IgG antibody versus 103 of 400 (25.8%, 95%CI: 21.5–30.3) HIV-uninfected pregnant women (*p*-value = 0.335) (Figure 2).

### 3.3. The Evolution of ZIKV IgG Seroprevalence during 1997–2017

The evolution of ZIKV IgG seroprevalence among all subjects over the five time periods was initially analyzed. In the period 2001–2003, 68 of 248 (27.4%, 95%CI: 22.0–33.4) subjects were ZIKV-IgG-positive. In the period 2008–2011, 25 of 102 subjects (24.5%, 95%CI: 16.5–34.0) were ZIKV-IgG-positive. In the period 2012–2014, 30 of 113 subjects (26.5%, 95%CI: 18.7–35.7) were ZIKV-IgG-positive. In the period 2015–2017, 66 of 399 subjects tested positive (16.5%, 95%CI: 13.0–20.6) for the ZIKV IgG antibody. ZIKV IgG seroprevalence was significantly lower during the period 2015-2017 as compared to other periods, likely as a result of the population enrolled during that period. Indeed, a large proportion of subjects were young men and non-pregnant women enrolled only in one city of northern Thailand.

When we restricted the analysis to pregnant women, ZIKV IgG seroprevalence looks stable over all the time periods: 24.3% in 1997–2000, 27.4% in 2001–2003, 24.5% in 2008–2011, 26.5% in 2021–2014 and 26.1% in 2015–2017. No significant differences were observed between periods (Figure 3).

### 3.4. Factors Associated with ZIKV IgG Seropositivity among Pregnant Women

In the univariable analysis, older age, being born or residing outside northern Thailand and a lower HIV-1 RNA load were significantly associated with ZIKV IgG positivity Table 2). In the multivariable analysis, factors found to be independently associated with ZIKV IgG positivity were older age (23–25 years versus 18–20 years: adjusted odd ratio (aOR) = 1.65, 95%CI: 1.03-2.63), being born outside northern Thailand (aOR = 1.95, 95%CI: 1.32-2.88) and lower HIV RNA (≤3.88 versus >3.88 log_10_ copies/mL: aOR = 1.46, 95%CI: 1.05-2.04).

## 4. Discussion

In the absence of systematic data collection on ZIKV infection over time in Thailand, public health measures to limit potential ZIKV outbreaks cannot be taken. This study assessed the ZIKV IgG seroprevalence in adults aged 18–25 years in Thailand from 1997 to 2017, which covers the period when outbreaks of ZIKV infection were reported. We found no association between HIV-infection status and ZIKV IgG positivity among pregnant women. ZIKV IgG seroprevalence in pregnant women was stable over this 20 year period, ranging from 24.3 to 27.4%. Older age, being born outside northern Thailand and a lower HIV-1 RNA load were found to be independently associated with ZIKV IgG positivity.

To the best of our knowledge, this is the first study assessing ZIKV IgG seroprevalence among young HIV-infected or HIV-uninfected pregnant women. We found similar ZIKV IgG positivity rates among these two groups, which may be due to the relatively preserved immunity in the HIV-infected women randomly selected in this study. Indeed, the median CD4+ T-cell count was 410 cells/mm^3^ (95% CI: 280–540), and none of the subjects living with HIV had severe clinical complications before enrolling in the original clinical studies. Our results also suggest that HIV infection may not have impaired the immune response to ZIKV.

Our study provides new indirect data on the circulation of ZIKV over the past two decades. ZIKV IgG seroprevalence was stable, ranging from 24.3 to 27.4% during 1997–2017. These results are consistent with the overall ZIKV IgG seroprevalence of 29% found in HIV- or HBV-infected pregnant women with a median age of 25.2 years in Thailand during 1997–2015 [23]. Our data of ZIKV IgG seroprevalence in the 1997–2000 period suggests that ZIKV was circulating in Thailand before 1997, which supports findings by Ponds et al. of ZIKV positive serology in Thailand since 1954 [12]. Another study using time-resolved phylogenetic tree analyses of ZIKV sequences obtained in Thailand also suggested a persistent circulation of ZIKV in Thailand since at least 2002, although this estimation was based on sequence data that were dated, at the earliest, in 2006 [24].

The risk of being ZIKV-IgG-positive was doubled in subjects who were born/living in regions outside northern Thailand as compared to those born/living in the northern region. When the region of enrollment was considered instead of region of birth in the multivariable analysis model, the same factors were found to be independently associated with ZIKV IgG positivity. ZIKV spread depends on various factors: mosquito vectors, environments for mosquitoes breeding and host behaviors, including people’s lifestyle and socioeconomic status. A possible hypothesis could be the different distribution of the mosquito vector and the variation in the environmental conditions needed for optimal mosquito breeding [25]. The ZIKV IgG seroprevalence during the last period, 2015–2017, was lower in the non-pregnant population as compared to the prevalence in the pregnant population. This may be explained by the fact that most of subjects were enrolled in Chiang Mai and living in urban areas. In northern Thailand, lower temperatures and humidity conditions may be less favorable for the spread of mosquitos [26]. The less favorable conditions for mosquito breeding in northern Thailand was shown in a survey study of the *Aedes* population using an Ovitrap to number the eggs laid by mosquitoes [27]. This study, conducted during 2012–2019 across 32 provinces of Thailand, showed the highest average eggs per trap and percent of *Aedes*-positive traps in the south, followed by the central, northeast and north regions, [27]. In addition, those living in cities may benefit from better mosquito control campaigns. This combination may contribute to an overall lower exposure to mosquitoes and, thus, to ZIKV.

Since the risk of exposure of an individual to mosquito bites increases with the age, the cumulative risk of infection is greater in older individuals. This is consistent with our finding that ZIKV IgG prevalence was higher among pregnant women aged 23–25 years compared to those aged 18–20 years (30.2% vs. 21.3%, *p* = 0.04). It is unclear why the prevalence of ZIKV IgG was higher among HIV-infected pregnant women with HIV RNA levels below the median. One hypothesis is that those individuals may have less inflammation and a better immunity. However, this warrants further confirmation.

Our study has some limitations. First, a high proportion of subjects were pregnant women infected with HIV or HBV [15,16,17,18,19,20], while the last period (2015–2017) samples were collected from a young population of male or female subjects seeking testing for HIV or other infections [21]. However, when we restricted the analysis to pregnant women only, the ZIKV IgG seroprevalence was stable. Second, as some clinical and socio-economic information was not available for analysis, further study is needed to confirm our findings and identify other potential confounding factors. Third, this study was conducted in endemic areas of the dengue virus. Thus, a cross-reactivity from the pre-existing anti-DENV IgG may have led to an overestimation of the ZIKV IgG seroprevalence. However, our ZIKV IgG seroprevalence results are consistent with the low exposure to ZIKV of healthy Thai people reported in 2017: 20% of 135 healthy subjects (95% CI: 14.0–28.2%) were positive for the ZIKV neutralizing antibody [28].

## 5. Conclusions

There was no evidence that the overall ZIKV IgG seroprevalence in populations aged 18–25 years in Thailand has evolved during 1997–2017, and it appeared to be stable at around 25%, suggesting that ZIKV has been circulating for more than 20 years. This study suggests that a large proportion of the population in Thailand probably remains susceptible to ZIKV infection, which could be the ground for future outbreaks affecting non-immune pregnant women with a potential for severe adverse pregnancy outcomes. Continued surveillance of the ZIKV spread in Thailand is needed to inform public health policies.

## Figures and Tables

**Figure 1 viruses-14-00368-f001:**
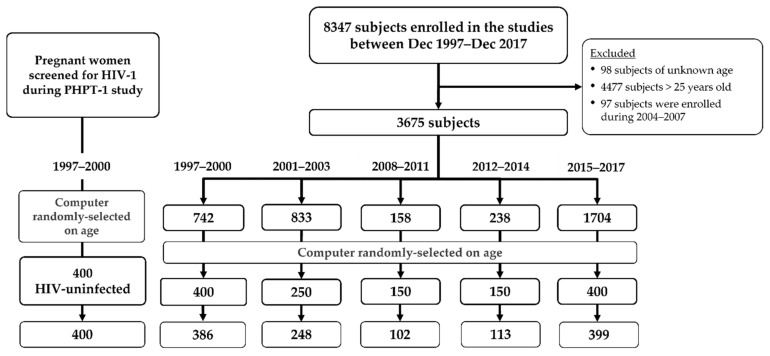
Study population: subjects were enrolled between 1997 and 2017, over five time periods: 1997–2000, 2001–2003, 2008–2011, 2012–2014 and 2015–2017. The target numbers of subjects randomly selected for each period were 400, 250, 150, 150 and 400, respectively. An additional group of 400 HIV-uninfected pregnant women enrolled in the 1997−1999 period was included. The bottom row indicates the number of selected subjects with the available samples.

**Figure 2 viruses-14-00368-f002:**
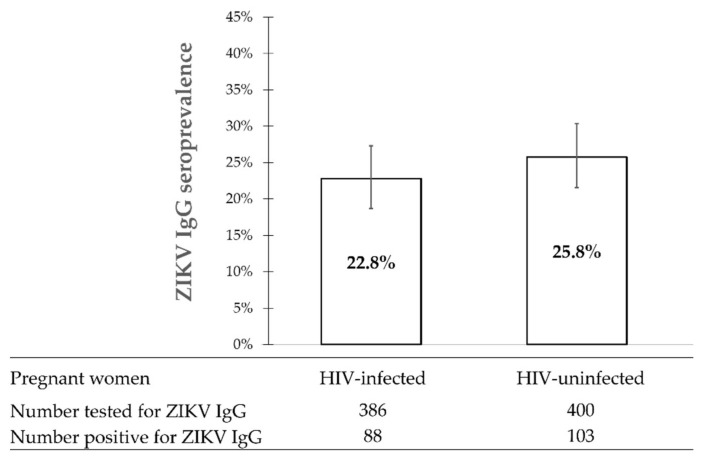
ZIKV IgG seroprevalence among pregnant women in Thailand according to HIV status during 1997–2000. The whisker error bars represent the 95% confidence intervals.

**Figure 3 viruses-14-00368-f003:**
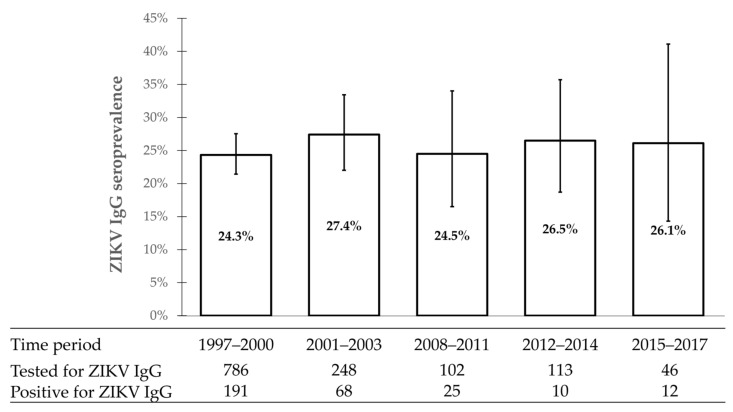
Evolution of ZIKV IgG seroprevalence among pregnant women in Thailand during 1997–2017. The whisker error bars represent the 95% confidence intervals.

**Table 1 viruses-14-00368-t001:** Characteristics of the study population.

Characteristics		Overall (n = 1648)		HIV-Uninfected Pregnant Women 1997–2000		Period 1997–2000		Period 2001–2003		Period 2008–2011		Period 2012–2015		Period 2015–2017	
		n/N or n	Percentage or Median (IQR)	n/N or n	Percentage or Median (IQR)	n/N or n	Percentage or Median (IQR)	n/N or n	Percentage or Median (IQR)	n/N or n	Percentage or Median (IQR)	n/N or n	Percentage or Median (IQR)	n/N or n	Percentage or Median (IQR)
Sex	Female	1464/1648	88.8	400/400	100.0	386/386	100.0	248/248	100.0	102/102	100.0	113/113	100.0	215/399	53.9
	Male	174/1648	10.6	0/400	0.0	0/386	0.0	0/248	0.0	0/102	0.0	0/113	0.0	174/399	43.6
	Other	10/1648	0.6	0/400	0.0	0/386	0.0	0/248	0.0	0/102	0.0	0/113	0.0	10/399	2.5
Age (years old)		1648	22.0 (20.0, 23.0)	372	22.0 (20.0, 23.0)	386	22.0 (20.1, 23.0)	248	22.0 (20.0, 23.0)	102	22.2 (20.2, 23.1)	113	21.9 (19.9, 23.7)	399	22.0 (20.0, 23.0)
Pregnancy (denominator: females)		1295/1464	88.5	400/400	100	386/386	100	248/248	100	102/102	100	113/113	100	46/215	21.4
Gestational age (weeks)		1262	25.0 (16.7, 29.7)	347	15.9 (11.3, 22.9)	386	21.4 (16.9, 25.3)	248	29.7 (28.0, 33.0)	102	32.4 (32.0, 33.7)	113	26.4 (20.6, 33.9)	46	28.1 (28.0, 28.6)
Region of birth	Central	250/1522	16.4	19/354	5.4	46/339	13.6	60/244	24.6	35/102	34.3	24/107	22.4	66/376	17.6
	Northern	757/1522	49.7	251/354	70.9	156/339	46.0	39/244	16	23/102	22.5	24/107	22.4	264/376	70.2
	Northeastern	173/1522	11.4	4/354	1.1	5/339	1.5	83/244	34	28/102	27.5	34/107	31.8	19/376	5.1
	Eastern	273/1522	17.9	78/354	22	125/339	36.9	46/244	18.9	6/102	5.9	13/107	12.1	5/376	1.3
	Western	25/1522	1.6	0/354	0.0	0/339	0.0	7/244	2.9	4/102	3.9	2/107	1.9	12/376	3.2
	Southern	38/1522	2.5	2/354	0.6	7/339	2.1	9/244	3.7	2/102	2.0	8/107	7.5	10/376	2.7
	Foreign country	6/1522	0.4	0/354	0.0	0/339	0.0	0/244	0	4/102	3.9	2/107	1.9	0/376	0.0
Region of enrollment	Central	250/1644	15.2	19/398	4.8	58/386	15.0	73/242	30.2	41/102	40.2	30/114	26.3	29/396	7.3
	Northern	868/1644	52.8	256/398	64.3	168/386	43.5	37/242	15.3	24/102	23.5	32/114	28.1	351/396	88.6
	Northeastern	70/1644	4.3	0/398	0.0	0/386	0.0	39/242	16.1	7/102	6.9	19/114	16.7	5/396	1.3
	Eastern	413/1644	25.1	121/398	30.4	152/386	39.4	81/242	33.5	27/102	26.5	22/114	19.3	10/396	2.5
	Western	7/1644	0.4	0/398	0.0	0/386	0.0	0/242	0.0	0/102	0.0	1/114	0.9	0/396	0.0
	Southern	36/1644	2.2	2/398	0.5	8/386	2.1	12/242	5.0	3/102	2.9	10/114	8.8	1/396	0.3
Education	Higher than bachelor’s degree	5/1638	0.3	0/396	0.0	0/383	0.0	0/248	0.0	0/102	0.0	0/113	0.0	5/396	1.3
	College/University	369/1638	22.5	21/396	5.3	25/383	6.5	25/248	10.1	14/102	13.7	16/113	14.2	268/396	67.7
	High school	188/1638	11.5	37/396	9.3	24/383	6.3	24/248	9.7	13/102	12.7	18/113	15.9	72/396	18.2
	Secondary school/Vocational certificate	394/1638	24.5	92/396	23.2	95/383	24.8	67/248	27.0	48/102	47.1	59/113	52.2	33/396	8.3
	Primary school	482/1638	29.4	179/396	45.2	164/383	42.8	88/248	35.5	24/102	23.5	15/113	13.3	12/396	3.0
	Lower than primary school	175/1638	10.7	55/396	13.9	75/383	19.6	36/248	14.5	3/102	2.9	5/113	4.4	1/396	0.3
	Others	25/1638	1.5	12/396	3.0	0/383	0.0	8/248	3.2	0/102	0.0	0/113	0.0	5/396	1.3
Marital status	Living with partner	834/892	93.5	n.a. ^1^	n.a. ^1^	372/384	96.9	224/247	90.7	91/102	89.2	102/113	90.3	45/46	97.8
	Divorced/Not living with partner/Widowed/Single	53/892	5.9	n.a. ^1^	n.a. ^1^	12/384	3.1	19/247	7.7	10/102	9.8	11/113	9.7	1/46	2.2
	Others	5/892	0.6	n.a. ^1^	n.a. ^1^	0/384	0.0	4/247	1.6	1/102	1.0	0/113	0.0	0/46	0.0
Number of household members	1 (Living alone)	101/611	16.5	n.a. ^1^	n.a. ^1^	n.a. ^1^	n.a. ^1^	n.a. ^1^	n.a. ^1^	3/102	2.9	0/112	0.0	98/397	24.7
	2 people	136/611	22.3	n.a. ^1^	n.a. ^1^	n.a. ^1^	n.a. ^1^	n.a. ^1^	n.a. ^1^	26/102	25.5	32/112	28.6	78/397	19.6
	3 people	96/611	15.7	n.a. ^1^	n.a. ^1^	n.a. ^1^	n.a. ^1^	n.a. ^1^	n.a. ^1^	15/102	14.7	16/112	14.3	65/397	16.4
	4 people	110/611	18	n.a. ^1^	n.a. ^1^	n.a. ^1^	n.a. ^1^	n.a. ^1^	n.a. ^1^	21/102	20.6	28/112	25	61/397	15.4
	More than 4 people	168/611	27.5	n.a. ^1^	n.a. ^1^	n.a. ^1^	n.a. ^1^	n.a. ^1^	n.a. ^1^	37/102	36.3	36/112	32.1	95/397	23.9
Multiple partner		77/236	32.6	n.a. ^1^	n.a. ^1^	n.a. ^1^	n.a. ^1^	n.a. ^1^	n.a. ^1^	n.a. ^1^	n.a. ^1^	n.a. ^1^	n.a. ^1^	77/236	32.6
Occupation	Unemployed or Housewife	487/1603	30.4	89/397	22.4	25/386	6.5	243/248	98.0	56/102	54.9	64/110	58.2	10/360	2.8
	Agriculturist/Fishery	176/1603	11	106/397	26.7	63/386	16.3	2/248	0.8	3/102	2.9	2/110	1.8	0/360	0.0
	Commercial/Private business/ Self-employed	128/1603	8.0	34/397	8.6	42/386	10.9	1/248	0.4	22/102	21.6	21/110	19.1	8/360	2.2
	Office man	152/1603	9.5	27/397	6.8	124/386	32.1	0/248	0.0	1/102	1.0	0/110	0.0	0/360	0.0
	Labor/Housekeeper	292/1603	18.2	131/397	33	121/386	31.3	0/248	0.0	12/102	11.8	19/110	17.3	9/360	2.5
	Student	303/1603	18.9	3/397	0.8	2/386	0.5	0/248	0.0	3/102	2.9	3/110	2.7	292/360	81.1
	Others	65/1603	4.1	7/397	1.8	9/386	2.3	2/248	0.8	5/102	4.9	1/110	0.9	41/360	11.4
Risk behavior	Alcohol consumption	290/404	71.8	n.a. ^1^	n.a. ^1^	24/24	100.0	16/16	100.0	7/7	100.0	9/9	100.0	234/348	67.2
	Smoking	59/349	16.9	n.a. ^1^	n.a. ^1^	n.a. ^1^	n.a. ^1^	n.a. ^1^	n.a. ^1^	n.a. ^1^	n.a. ^1^	n.a. ^1^	n.a. ^1^	59/349	16.9
	Drug use	73/351	20.8	n.a. ^1^	n.a. ^1^	n.a. ^1^	n.a. ^1^	n.a. ^1^	n.a. ^1^	n.a. ^1^	n.a. ^1^	n.a. ^1^	n.a. ^1^	73/351	20.8
	Any of these	301/407	74	n.a. ^1^	n.a. ^1^	24/24	100.0	16/16	100.0	7/7	100.0	9/9	100.0	245/351	69.8
Infection status	Anti-HIV positive	844/1645	51.2	- ^2^	- ^2^	386/386	100.0	247/248	99.6	101/101	100.0	106/112	94.6	4/398	1.0
	HIV RNA load (log_10_ copies/mL)	838	3.88 (3.21, 4.46)	- ^2^	- ^2^	386	3.92 (3.32, 4.40)	246	4.0 (3.33, 4.70)	102	3.57 (2.16, 4.19)	99	3.81 (4.52, 3.04)	5	4.88 (3.9, 5.01)
	HIV RNA load among pregnant women (log_10_ copies/mL)	834	3.87 (3.21, 4.45)	- ^2^	- ^2^	386	3.92 (3.32, 4.40)	246	4.0 (3.33, 4.70)	102	3.57 (2.16, 4.19)	99	3.81 (4.52, 3.04)	- ^2^	- ^2^
	HBsAg positive	111/1245	8.9	n.a. ^1^	n.a. ^1^	28/385	7.3	16/246	6.5	5/102	4.9	8/113	7.1	54/399	13.5
	Anti-HCV positive	22/1246	1.8	n.a. ^1^	n.a. ^1^	13/384	3.4	3/248	1.2	5/105	4.9	0/113	0.0	1/399	0.3
	Syphilis positive	3/353	0.8	n.a. ^1^	n.a. ^1^	n.a. ^1^	n.a. ^1^	n.a. ^1^	n.a. ^1^	n.a. ^1^	n.a. ^1^	n.a. ^1^	n.a. ^1^	3/353	0.8
Blood chemistry testing	Fasting blood sugar (mg/dL)	112	82 (73, 91)	n.a. ^1^	n.a. ^1^	17	90 (85, 109)	7	88.8 (83, 94)	42	78.5 (71, 84)	43	77 (71, 84)	3	105 (68, 120)
	Cholesterol (mg/dL)	248	217 (180, 258.5)	n.a. ^1^	n.a. ^1^	32	197 (160.5, 225)	38	173.5 (152, 206)	102	244 (217, 280)	106	211.5 (182, 252)	6	200 (179, 220)
	AST (IU/L)	184	21.0 (17, 29.5)	n.a. ^1^	n.a. ^1^	32	30.5 (21.5, 46)	32	22(18, 34.5)	40	20.5 (16.5, 31.5)	34	20 (16, 24)	46	19 (16, 22)
	ALT (IU/L)	866	14.0 (10.0, 20.0)	n.a. ^1^	n.a. ^1^	385	14 (10, 20)	239	16(11,15)	102	14 (10, 21)	111	12 (9, 15)	46	17.5 (12, 20)
Hematological testing	Hemoglobin (g/dL)	893	10.8 (11.6, 10)	n.a. ^1^	n.a. ^1^	384	10.6 (9.9, 11.4)	248	11 (10.1, 11.65)	102	10.9 (10.2, 11.6)	113	11 (10.2, 11.6)	46	11.35 (10.7, 12.0)
	Hematocrit (%)	895	33.0 (35.0, 30.9)	n.a. ^1^	n.a. ^1^	386	33.0 (30.9, 35.1)	248	32.9 (30.55, 34.85)	102	32.7 (31.0, 34.4)	113	32.9 (30.8, 35.0)	46	33.9 (32.0, 35.6)
	RBC count (million cells/mL)	516	4.0 (3.6, 4.4)	n.a. ^1^	n.a. ^1^	129	3.97 (3.54, 4.40)	127	4.33 (3.95, 4.72)	102	3.71 (3.49, 3.95)	112	3.94 (3.61, 4.26)	46	4.15 (3.80, 4.47)
	Platelet count (thousand/mm^3^)	596	78.5 (180.0, 258.5)	n.a. ^1^	n.a. ^1^	87	241 (197, 286)	248	271 (230.5, 318.5)	102	271.5 (233, 328)	113	255 (217, 298)	46	254.5 (219, 283)
	WBC count (cells/mm^3^)	877	8880 (10,600, 6400)	n.a. ^1^	n.a. ^1^	386	8800 (7300, 10,700)	248	8525 (7400, 10,700)	102	8780 (7650, 10,490)	113	9000(7700, 10,120)	46	11,150 (9100, 12,710)
	Absolute CD4 T-cell (cells/mm^3^)	814	410 (280, 550)	- ^2^	- ^2^	358	378.5 (250, 540)	248	405.5 (266.5, 541)	102	518.5 (413, 654)	102	394.5(292, 516)	4	565 (417, 853)
	Absolute CD4 T-cell among pregnant women (cells/mm^3^)	810	409.5 (280, 550)	- ^2^	- ^2^	358	378.5 (250, 540)	248	405.5 (266.5, 541)	102	518.5 (413, 654)	102	394.5(292, 516)	n.a. ^1^	n.a. ^1^

Note: ^1^ Not available; ^2^ Not applicable.

**Table 2 viruses-14-00368-t002:** Factors associated with ZIKV IgG positivity among pregnant women (N = 1295).

Characteristics		n/N	%	Univariate Analysis		Multivariate Analysis	
				Odds Ratio (95%CI)	*p*	Adjusted Odds Ratio (95%CI)	*p*
Period	1997–2000	88/386	22.8	1			
	2001–2003	68/248	27.4	1.28 (0.89–1.85)	0.19		
	2008–2011	25/102	24.5	1.10 (0.66–1.83)	0.72		
	2012–2014	30/113	26.5	1.22 (0.76–1.98)	0.41		
	2015–2017	12/46	26.1	1.20 (0.59–2.41)	0.62		
Age	18–20 years	47/221	21.3	1		1	
	>20–22 years	51/226	22.6	1.08 (0.69–1.69)	0.74	1.06 (0.65–1.73)	0.81
	>22–23 years	55/216	25.5	1.26 (0.81–1.97)	0.30	1.36 (0.85–2.20)	0.20
	>23–25 years	70/232	30.2	1.60 (1.04–2.45)	*0.03*	1.65 (1.03–2.63)	0.04
Gestational age (N = 1262)	1–13 weeks	11/46	23.9	1			
	>13–28 weeks	111/466	23.8	0.99 (0.49–2.02)	0.99		
	>28 weeks	101/383	26.4	1.14 (0.56–2.33)	0.72		
Region of birth (N = 1177)	North	44/254	17.3	1		1	
	Other	168/569	29.5	2.00 (1.38–2.90)	<0.001	1.95 (1.32–2.88)	<0.001
Region of enrollment (N = 1293)	North	45/277	16.3	1			
	Other	178/618	28.8	2.09 (1.45–3.00)	<0.001	- ^1^	n.i. ^2^
Education (N = 1288)	Lower than secondary school	99/421	23.5	1			
	Secondary school/Vocational certificate	76/283	26.9	1.19 (0.84–1.69)	0.32		
	Higher than secondary school	46/177	26	1.14 (0.76–1.71)	0.52		
	Other	2/11	18.2	0.72 (0.15–3.40)	0.68		
Marital status (N = 892)	Divorced/Not living with partner/Widowed/Singer	18/53	34	1.62 (0.90–2.92)	0.11	1.45 (0.77–2.75)	n.s.^3^
	Living with partner	201/834	24.1	1		1	
	Other	2/5	40	2.10 (0.35–12.65)	0.42	1.98 (0.33–12.09)	n.s.^3^
HIV status (N = 1292)	HIV negative	118/452	26.1	1.24 (0.67–2.30)	0.50		
	HIV positive	207/840	24.6	1			
HIV-1 RNA load (N = 834)	≤3.88 log _10_ copies/mL	120/420	28.6	1.55 (1.13–2.13)	0.01	1.46 (1.05–2.04)	0.03
	>3.88 log _10_ copies/mL	85/414	20.5	1		1	

Note: ^1^ Not applicable, ^2^ not included due to collinearity with region of birth, ^3^ not significant.

## Data Availability

Not acceptable.

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
