# Peer review of "Zika Virus Immunoglobulin G Seroprevalence among Young Adults Living with HIV or without HIV in Thailand from 1997 to 2017"

_viruses, 2022, doi:10.3390/v14020368_

Round 1
Reviewer 1 Report
This manuscript by Choyrum et al. seeks to understand patterns of seroprevalence to Zika virus over the course of 20 years in Thailand. The authors accomplish this well, with a solid approach that provides valuable public health information such as how HIV viral loads and region of birth impact ZIKV seropositivity. The article is clear, relevant and well-structured, with easy to understand results and good interpretation. My comments are minor and easily addressed via amendments to the text.
- 1: It is unclear why 97 subjects were excluded from the 2004-2007 enrollment period. The sentence where this is explained needs clarification.
- More information about the ELISA should be included, especially the product number.
- 3: Were any significant differences noted here? Is the 2015-2017 period significantly lower than other periods?
- Line 217: This should be Chiang Mai I believe.
Author Response
Reviewer #1: Comments and Suggestions for Authors
This manuscript by Choyrum et al. seeks to understand patterns of seroprevalence to Zika virus over the course of 20 years in Thailand. The authors accomplish this well, with a solid approach that provides valuable public health information such as how HIV viral loads and region of birth impact ZIKV seropositivity. The article is clear, relevant and well-structured, with easy-to-understand results and good interpretation. My comments are minor and easily addressed via amendments to the text.
1) It is unclear why 97 subjects were excluded from the 2004-2007 enrollment period. The sentence where this is explained needs clarification.
Response: We agree with the reviewer that the information provided in the Figure 1 legend was not clear. We added to the materials and methods section (study population; Lines 93-96) a sentence explaining the reason “why 97 subjects were excluded from the 2004-2007 enrollment period”. The sentence reads as follows:
“...Only 97 women enrolled in the perinatal HIV prevention studies conducted during the period 2004-2007. These 97 pregnant women were not included in this Zika study since their number was too low to allow for an appropriate random age-based selection. …”
2) More information about the ELISA should be included, especially the product number.
Response: Following this advice, we added information about the ELISA test kit in the Materials and methods section (laboratory testing; Lines 115-116). The sentence reads as follows:
“…using an indirect ELISA test (EUROIMMUN Anti-Zika Virus IgG ELISA, Product number: EI 2668-9601 G, Lübeck, Germany...”
3) Were any significant differences noted here? Is the 2015-2017 period significantly lower than other periods?
Response: ZIKV IgG seroprevalence was significantly lower during the period 2015-2017 as compared to other periods likely as a result of the population enrolled during that period. Indeed, a large proportion of subjects were young men and non-pregnant women enrolled only in one city of Northern Thailand. We added these sentences in the result section (Evolution of ZIKV IgG seroprevalence during 1997–2017; Line 172-176).
4) Line 217: This should be Chiang Mai I believe.
Response: This typo has been corrected (Line 232).

Reviewer 2 Report
The paper from Choyrum and colleagues is a retrospective report of the seroprevalence of ZIKV IgG antibodies in a series of cohorts
of subjects from Thailand.
The introduction is brief, but provides the useful informations to trace the epidemiological situtation in the country during the timeframe
of the study, but would benefit form a short paragraph of general virological information about the virus itself.
The methods are sufficiently well detailed.
For the results, table 1 is very interesting, but it is not very clear how were the subjects subdivided in the presented groups: the authors should,
in the main text, explain which sub-classifications they performed for allocating the subjects in the 3 groups shown in table along with the 'overall',
and why they choose this subdivion among others (like, for example, the division in non-overlapping groups based on a peculiar condition, or instead a simple
division based on the original cohort of the subjects). Apart from this, the organization in paragraphs could be optimized, since for example par 3.2 consists in
only one figure, while par 3.3 and par 3.4 should absolutely be merged, since they present the same data differing only on one point (and, to be honest, fig 3 should
not presented, because the point for period 2015-17 is biased by the completely different cohort composition, which is corrected in fig 4)
So, the authors could merge 3.3 and 3.4 and merge fig 3 and 4 or eliminate fig 3.
The dicussion and conclusions are coherent with the results, even though some more efforts to explain the findings, for example the strong correlation with geographical origins, would be
welcome (are there any data available on mosquito distribution in the different regions?)
Some minor points:
- line 51: 'exhibits'
- line 88: in fig.1 please explain the PHPT acronym.
- line 89: the caption should provide a synthetic explanation of the figure.
- line 92: 'before any antiretroviral treatment'
- line 95: 'and a test was considered..'
- line 100: 'years in which'
- line 142: fig1: why is there a group named 'all pregnant women' wih N=895, while the main text says that the pregnant women enrolled were 1295? Which number is incorrect?
- line 143: 'were subtracted'
- line 179: the multivariate odd aratio for 'region of enrollment' is not shown due to 'collinearity with the region of birth'. Actually, since the ZIKV exposure is in
function of the region in which people lives and not in which they were born, it would be logically more correct to show this value and exclude the one for region of birth.
Author Response
Reviewer #2:
Open Review
Comments and Suggestions for Authors
The paper from Choyrum and colleagues is a retrospective report of the seroprevalence of ZIKV IgG antibodies in a series of cohorts of subjects from Thailand.
The introduction is brief, but provides the useful informations to trace the epidemiological situation in the country during the timeframe of the study, but would benefit form a short paragraph of general virological information about the virus itself.
Response: We thank the reviewer for the suggestion and added two sentences about the virology of ZIKV at the beginning of the introduction (Lines 48-52). The new sentences read as follows:
“Zika virus (ZIKV) is an enveloped, positive single-strand RNA virus belonging to Flaviviridae family, Flavivirus genus. The RNA genome is composed of about 10,800 nucleotides encoding three structural proteins capsid, precursor membrane and envelope protein and seven nonstructural (NS) proteins i.e., NS1, NS2A, NS2B, NS3, NS4A, NS4B and NS5 [1]…”
The methods are sufficiently well detailed. For the results, table 1 is very interesting, but it is not very clear how were the subjects subdivided in the presented groups: the authors should, in the main text, explain which sub-classifications they performed for allocating the subjects in the 3 groups shown in table along with the 'overall', and why they choose this subdivion among others (like, for example, the division in non-overlapping groups based on a peculiar condition, or instead a simple division based on the original cohort of the subjects).
Response: We thank the reviewer for picking this point. We agree that we should have presented the characteristics of subjects in each time period. We initially prepared a table with the characteristics of study population by time periods, but considered that this table was not easy to read due to the number of columns. We have now replaced the current Table 1 with the table that describes subject’s characteristic by time periods. The new table is provided at the end of our responses to the reviewers.
Apart from this, the organization in paragraphs could be optimized, since for example par 3.2 consists in only one figure, while par 3.3 and par 3.4 should absolutely be merged, since they present the same data differing only on one point (and, to be honest, fig 3 should not presented, because the point for period 2015-17 is biased by the completely different cohort composition, which is corrected in fig 4). So, the authors could merge 3.3 and 3.4 and merge fig 3 and 4 or eliminate fig 3.
Response: We agree with the reviewer and have re-organized the result section as follows: the previous figure 3 was replaced with figure 4 (Line 183-184) and the section 3.3 and 3.4 have been merged (Line 167-181).
The discussion and conclusions are coherent with the results, even though some more efforts to explain the findings, for example the strong correlation with geographical origins, would be welcome (are there any data available on mosquito distribution in the different regions?)
Response: We thank the reviewer for this comment and added more information on the mosquito distribution in the different regions of Thailand. The following paragraph has been added to the discussion section, Fourth paragraph, Line 234-239.
“The less favorable conditions for mosquitos breeding in Northern Thailand has been showed in a survey study of Aedes population using an Ovitrap to number the eggs laid by mosquitoes [27]. This study, conducted during 2012-2019 across 32 provinces of Thailand, showed the highest average eggs per trap and percent of Aedes positive trap in the South followed by Central, Northeast and North regions, respectively [27].”
Some minor points:
- Line 51: 'exhibits'
Response: This has been corrected (Line 55)
- Line 88: in fig.1 please explain the PHPT acronym.
Response: The PHPT research collaboration initially focused on the prevention of mother-to-child transmission of HIV. It has evolved over the past 15 years to conduct research to improve treatment for children and adults living with HIV and other viral co-infections, and more recently prevention of perinatal transmission of hepatitis B virus. For this reason, the initial acronym is not anymore relevant and the research collaboration is known as PHPT.
- Line 89: the caption should provide a synthetic explanation of the figure.
Response: We agree with the reviewer that a synthetic explanation of the figure 1 is needed. The Figure 1 legend reads as follows (Line 108-112):
“Study population: subjects were enrolled between 1997 and 2017, over five time periods: 1997–2000, 2001–2003, 2008–2011, 2012–2014, and 2015–2017. The target number of subjects randomly selected at each period was 400, 250, 150, 150, and 400, respectively. An additional group of 400 HIV-uninfected pregnant women enrolled in the 1997-1999 period was included. The bottom row indicates the number of selected subjects with available samples.”
- Line 92: 'before any antiretroviral treatment'
Response: This has been corrected (Line 114)
- Line 95: 'and a test was considered..'
Response: The paragraph of this phase has been modified and the phase has been corrected (Line 118)
- Line 100: 'years in which'
Response: The paragraph of this phase has been moved into Materials and methods (study population) section and the phase has been corrected (Line 97)
- Line 142: fig1: why is there a group named 'all pregnant women' with N=895, while the main text says that the pregnant women enrolled were 1295? Which number is incorrect?
Response: 1,295 is the total number of pregnant women included in the analysis, while 895 is the number of pregnant women excluding the number of HIV-uninfected pregnant women during 1997-2000. We have replaced the previous Table 1 where populations were overlapping with a table that describes the characteristics of subjects by time periods. However, the previous table is now provided as a supplementary table.
- Line 143: 'were subtracted'
Response: The sentence of this phase has been deleted (Line 157)
- Line 179: the multivariate odd ratio for 'region of enrollment' is not shown due to 'collinearity with the region of birth'. Actually, since the ZIKV exposure is in function of the region in which people lives and not in which they were born, it would be logically more correct to show this value and exclude the one for region of birth.
Response: We agree with the reviewer that it would be more logical to keep the region of enrollment over the region of birth. However, we decided to keep the region of birth because a large majority of young people in Thailand continue to reside in areas where they were born and have spent their childhood.
When the region of enrollment was considered in the multivariable analysis, it remained independently associated with ZIKV IgG positivity (aOR = 2.00, 95%CI: 1.36-2.93). We found the same other factors independently associated with Zika IgG positivity, i.e. an older age [23–25 years versus 18–20 years: adjusted odd ratio (aOR) = 1.78, 95%CI: 1.13-2.81] and a lower HIV RNA (≤3.88 versus >3.88 log10 copies/mL: aOR = 1.50, 95%CI: 1.09-2.08).
We added the following sentence into the discussion section (Line 224-226)
“…When the region of enrollment was considered instead of region of birth in the multivariable analysis model, the same factors were found independently associated with ZIKV IgG positivity. …”
*********************************************************

Reviewer 3 Report
The authors of the manuscript “Zika virus immunoglobulin G seroprevalence among young adults living with HIV or without HIV in Thailand from 1997 to 2017” analyzed retrospectively samples for ZIKV IgG detection in young adults in Thailand. The presented data is relevant; however, this reviewer encourages the authors to provide:
- Verification of the total number of 8,347 subjects enrolled in the study. According with the ClinicalTrial.org website, in the trials: NCT00386230, NCT00398684, NCT00142337, NCT00409591, NCT01745822, NCT02752152 were enrolled 1554, 1792, 244, 435, 654, and 1961 participants respectively with total of 6640 participants in the 6 clinical trials.
- Laboratory testing: indicate details about quality control for laboratory serology detection.
Author Response
Reviewer #3:
Comments and Suggestions for Authors
The authors of the manuscript “Zika virus immunoglobulin G seroprevalence among young adults living with HIV or without HIV in Thailand from 1997 to 2017” analyzed retrospectively samples for ZIKV IgG detection in young adults in Thailand. The presented data is relevant; however, this reviewer encourages the authors to provide:
Verification of the total number of 8,347 subjects enrolled in the study. According with the ClinicalTrial.org website, in the trials: NCT00386230, NCT00398684, NCT00142337, NCT00409591, NCT01745822, NCT02752152 were enrolled 1554, 1792, 244, 435, 654, and 1961 participants respectively with total of 6640 participants in the 6 clinical trials.
Response: The number of participants in our Dataset was the actual number of participants coming at enrollment visit in any studies. There are several reasons for the discrepant numbers of participants between our Dataset and the numbers in ClinicalTrial.org website. First, for the earliest studies i.e. PHPT-1 and PHPT-2, the numbers of participants in ClinicalTrial.org website were the numbers of subjects expected to enroll based on the protocol and not the accrual number of participants. The ClinicalTrial.org website has evolved and for more recent studies, the number of participants is updated to include the accrual of participants based on the publications. Second, at the time we conducted this ZIKA IgG study, the iTAP and Napneung clinical studies were ongoing and the total numbers of enrollment were not updated on the ClinicalTrial.org website. We forgot to mention the PHPT-5 phase II study among the study conducted and have updated the reference (Line 84 and 86) and the NCT number of the study.
The table below shows the number of participants by study.
|
Project |
Number of participants in our Dataset |
NCT# |
Number of participants in ClinicalTrial.org website |
|
PHPT-1 |
1,437 |
NCT00386230 |
1,554 |
|
PHPT-2 |
2,028 |
NCT00398684 |
1,792 |
|
PHPT-4 |
257 |
NCT00142337 |
244 |
|
PHPT-5 phase I |
436 |
NCT00409591 |
435 |
|
PHPT-5 phase II |
391 |
NCT01511237 |
379 |
|
iTAP |
331 |
NCT01745822 |
654 |
|
Napneung |
3,467 |
NCT02752152 |
1,961 |
Laboratory testing: indicate details about quality control for laboratory serology detection.
Response: Following this advice, we added the following sentence in the Materials and methods section (Laboratory testing; Lines 117-118):
“…Each test run was validated with the kit positive and negative controls as internal controls. ….”
